# Empowering Women in Tobacco Control: A Participatory Study on Household Smoking Behavior in Aceh, Indonesia

**DOI:** 10.3390/ijerph22101490

**Published:** 2025-09-26

**Authors:** Hasrizal Saffutra, Mustanir Yahya, Rizanna Rosemary, Rosaria Indah, Dedy Syahrizal

**Affiliations:** 1Doctoral Program, Faculty of Medicine, Universitas Syiah Kuala, Banda Aceh 23111, Indonesia; hasrizal.ditra83@gmail.com; 2Department of Chemistry, Faculty of Science, Universitas Syiah Kuala, Banda Aceh 23111, Indonesia; 3Department of Communication Studies, Faculty of Social and Political Sciences, Universitas Syiah Kuala, Banda Aceh 23111, Indonesia; rizanna.rosemary@usk.ac.id; 4Department of Medical Education, Faculty of Medicine, Universitas Syiah Kuala, Banda Aceh 23111, Indonesia; rosariaindah@usk.ac.id; 5Department of Biochemistry, Faculty of Medicine, Universitas Syiah Kuala, Banda Aceh 23111, Indonesia; dedysyahrizal@usk.ac.id

**Keywords:** smoking cessation, women’s role, family health, participatory action research, qualitative research, Aceh, Indonesia

## Abstract

Tobacco smoking remains a critical public health concern in Indonesia, particularly in Aceh Province, where male smoking prevalence reaches 56.12%. Cultural permissiveness and weak enforcement of tobacco control regulations have contributed to this high prevalence. Women, especially wives, are central figures in family health and may play an essential role in influencing household smoking behavior. However, their roles and strategies remain underexplored, especially in conservative cultural settings. This qualitative study employed a Participatory Action Research (PAR) approach to examine the roles of women in controlling the smoking behavior of family members in Aceh. A total of 75 research subjects were selected from three districts (Aceh Singkil, Gayo Lues, and Pidie Jaya) using a combination of purposive sampling and snowball sampling methods. Data were collected through semi-structured in-depth interviews and were analyzed thematically using NVivo 15 software. The analysis framework was based on Lawrence Green’s PRECEDE model, which includes predisposing, enabling, and reinforcing factors. This study found that women demonstrated progressive attitudes toward smoking, evolving from passive acceptance to active responsibility. They employed both persuasive strategies (health education, emotional appeals, and motivation) and coercive actions (household smoking bans, threats, and withdrawal of privileges). Women also positioned themselves as health monitors and guardians within the household. Despite cultural limitations and gender hierarchy, many participants reported partial or complete success in encouraging their husbands to quit smoking, particularly when supported by religious norms and health awareness. Women play a pivotal role in shaping smoking-related behavior in the family. Empowering women through participatory frameworks and culturally tailored interventions can enhance their effectiveness as health advocates. This study underscores the need to integrate gender-sensitive strategies into national tobacco control policies, especially in culturally conservative regions.

## 1. Introduction

Tobacco use is one of the leading preventable causes of morbidity and mortality worldwide. According to the World Health Organization (WHO), tobacco kills more than 8 million people each year, including 1.2 million non-smokers exposed to secondhand smoke [1]. Despite global efforts to reduce tobacco consumption through regulatory policies, taxation, and public education campaigns, smoking remains prevalent in many low- and middle-income countries, particularly among men [2].

Indonesia ranks among the countries with the highest smoking prevalence in Southeast Asia. The 2018 Basic Health Research (Riskesdas) reported that 28.9% of the Indonesian population smokes, with 55.8% of adult males being active smokers [3]. In the province of Aceh, the situation is even more concerning, with male smoking prevalence reaching 56.12%, placing it among the highest in the country [4]. Cultural acceptance, weak enforcement of anti-smoking policies, and local norms that perceive smoking as part of male identity have all contributed to this public health challenge [5].

While tobacco control policies often focus on legislative action and mass media campaigns, less attention has been paid to the role of the family unit, especially women, in influencing smoking behavior at the household level [6]. Women, particularly wives and mothers, play a critical role in shaping family health practices. Their involvement in caregiving, health promotion, and behavioral monitoring positions them as potential agents of change in tobacco cessation efforts [7]. However, in patriarchal and culturally conservative societies such as Aceh, gender power dynamics, religious norms, and social expectations often limit women’s ability to intervene effectively in male-dominated behaviors such as smoking [8].

Several studies have examined social and environmental factors influencing smoking behavior. Still, there remains a lack of research exploring the active agency of women in tobacco control within traditional family systems [9]. The current literature does not sufficiently capture the strategies, constraints, and coping mechanisms employed by women who seek to reduce smoking behaviors in their households [10]. Furthermore, there is limited application of participatory frameworks that empower women not only as caregivers but also as community-based health advocates.

To address this gap, the present study explores the role of women in controlling the smoking behavior of family members in Aceh using a Participatory Action Research (PAR) approach. This method enables the co-construction of knowledge and community-led strategies grounded in local values and lived experiences. By doing so, this study aims to contribute to the development of more inclusive and culturally sensitive tobacco control interventions in Indonesia and similar contexts. This study makes a significant contribution to the tobacco control literature in low- and middle-income countries (LMICs) by offering a contextually grounded, community-based perspective on how women in culturally and religiously conservative regions such as Aceh, Indonesia, develop household-level smoking cessation strategies. Unlike most studies that focus on macro-level policies or quantitative data, this research fills a significant knowledge gap by presenting first-hand narratives of subjective experiences and relational dynamics that shape tobacco control within the domestic sphere.

### 1.1. Literature Review

#### 1.1.1. Tobacco Use and Family Dynamics

Tobacco consumption remains a significant global public health issue, causing more than 8 million deaths annually [11]. In low- and middle-income countries, where smoking prevalence is higher among men, exposure to secondhand smoke often disproportionately affects women and children in the household [12]. The domestic setting becomes a critical environment for both exposure and intervention. Studies have highlighted the significant role of family dynamics in influencing smoking behaviors [13]. Family support, particularly from spouses, has a measurable impact on smoking cessation outcomes. Their research indicates that when family members actively support quitting efforts, the likelihood of successful cessation increases [14]. This underscores the necessity to explore the interpersonal and household context of smoking behavior beyond the individual smoker.

#### 1.1.2. The Role of Women in Health Advocacy

Women are typically regarded as central health managers within the family, a role consistently observed in both global health discourse and local contexts, such as Indonesia, including Aceh. This positioning reflects their everyday responsibilities in ensuring household nutrition, caregiving, and health-related decision making. As caregivers, mothers, and spouses, women hold substantial influence over health-related decisions, including diet, healthcare utilization, and lifestyle habits such as smoking [15]. In conservative or patriarchal societies, such as Aceh, this influence is often culturally embedded, although not always explicitly acknowledged [16]. Previous studies have demonstrated the informal yet powerful authority of women in promoting smoke-free environments at home [17]. For instance, preview research documented how women households used social persuasion and emotional appeals to alter husbands’ smoking behaviors [18]. Similar dynamics are seen in Southeast Asia, where women act as mediators between public health messages and family-level behavioral change [19].

#### 1.1.3. Smoking Cessation and Family-Based Interventions

Effective smoking cessation often involves more than just individual willpower; it is also shaped by the environment in which the smoker resides. Family-based interventions have gained recognition for their potential to enhance the success of smoking cessation programs [20]. The social pressure and emotional attachment within families were more influential than public health campaigns alone [21]. Furthermore, interventions that include family education, particularly those targeting wives and mothers, have demonstrated improved outcomes in terms of smoking motivation and relapse prevention [22]. Such findings emphasize the need to empower non-smoking family members, especially women, as key actors in cessation programs.

#### 1.1.4. The PRECEDE Model as a Behavioral Framework

The PRECEDE-PROCEED model developed serves as a strategic planning framework for health promotion and behavioral interventions [23]. The PRECEDE component (Predisposing, Reinforcing, and Enabling Constructs in Educational Diagnosis and Evaluation) is particularly relevant to this study, as it offers a systematic approach to identifying behavioral determinants [24]. Predisposing factors include knowledge, beliefs, values, and attitudes that may influence readiness to change. Enabling factors refer to conditions that facilitate or hinder the change process, such as access to resources or services [25]. Reinforcing factors include social support or feedback that sustains behavioral change, such as encouragement from religious leaders or local regulations. Numerous studies that have applied the PRECEDE model to tobacco control have found it effective in identifying barriers and mobilizing community-based strategies [26]. Its flexibility allows for adaptation to cultural settings, making it suitable for application in Aceh.

#### 1.1.5. Participatory Action Research (PAR) and Community Engagement

PAR is a collaborative research approach that actively involves community members in identifying problems and developing solutions [27]. It is grounded in empowerment and co-creation of knowledge, particularly relevant when addressing behavior deeply rooted in culture and social norms [28]. Preview research argues that PAR facilitates genuine community ownership of health programs, which leads to greater sustainability and relevance [29]. In tobacco control, PAR enables women to share their experiences and influence interventions that reflect their realities [30]. The other researcher demonstrated that involving women in designing smoke-free campaigns in low-income settings led to higher compliance and engagement [31]. In the context of Aceh, PAR supports the inclusion of women not only as study participants but also as agents of transformation, bridging public health goals with cultural realities.

### 1.2. Conceptual Framework of Study

This study proposes an emergent conceptual framework based on the thematic findings described above. The framework illustrates the mechanisms through which women influence smoking behavior in the household and redefine gendered roles in health promotion. Figure 1 highlights how women act as central agents in promoting family health by influencing behaviors related to smoking through a combination of internal motivation and external support. Their role is shaped by enabling factors, such as access to health services, economic capacity, and education, as well as by reinforcing factors like support from religious leaders and encouragement from family. When these elements align, they facilitate behavior change and lead to a decrease in smoking among family members, underscoring women’s strategic position in health advocacy within households.

### 1.3. Research Questions

To understand the sociocultural dynamics and participatory role of women in promoting smoking cessation within Acehnese households, this study was guided by a series of research questions rooted in the PRECEDE model (Predisposing, Enabling, Reinforcing factors). These questions were designed to explore not only individual behavior but also structural and cultural influences that shape women’s capacity as health advocates in their families. The Participatory Action Research (PAR) framework enabled researchers and participants to collaboratively examine and reflect on women who demonstrated the ability to intervene directly in their household’s smoking behavior by implementing smoke-free rules, placing visual reminders (e.g., no-smoking signs), engaging in emotional dialogue with their husbands, and leveraging religious and community figures to reinforce anti-smoking messages. The following research questions were formulated:What roles do women play in influencing the smoking cessation behavior of their family members in Acehnese households?What are the predisposing factors (e.g., knowledge, attitudes, and intentions) that affect women’s involvement in supporting smoking cessation within the family?How do enabling factors (e.g., access to health services, economic conditions, and health education) facilitate or hinder women’s efforts in promoting smoke-free home environments?What reinforcing factors (e.g., religious values, family support, and local regulations) contribute to the effectiveness of women’s advocacy in controlling smoking behavior at home?How do women perceive their roles and responsibilities as health advocates within their households and communities concerning smoking behavior?

## 2. Materials and Methods

### 2.1. Study Design

This study employed an independent, exploratory qualitative design, guided by the Participatory Action Research (PAR) framework and the PRECEDE-PROCEED model [32]. Rather than implementing structured interventions, this study examined the cultural, social, and behavioral factors that influence women’s roles in household smoking cessation. Emphasis was placed on the co-construction of knowledge, where women and their families collaboratively developed practical strategies such as creating smoke-free zones, setting no-smoking times (e.g., during meals or around children), and displaying handwritten reminders. These strategies emerged through mutual dialogue and negotiation, reflecting women’s growing agency in health-related household decisions. Data were collected through two rounds of semi-structured interviews conducted at participants’ homes between June 2023 and March 2024. The first session gathered initial narratives, while the second session validated transcripts and confirmed emerging themes, thereby enhancing the trustworthiness of the findings. This participatory, iterative method underscores the importance of lived experiences in shaping culturally sensitive health advocacy and promoting behavioral change.

In line with the Participatory Action Research (PAR) framework, several participatory tools were employed to ensure that participants were not only sources of data but also active co-constructors of knowledge. These included the following:Iterative reflective interviews, conducted in two rounds, allowing women to revisit, refine, and validate their narratives.Photovoice exercises, where participants captured household contexts (e.g., children’s exposure to smoking, family routines) using mobile phone photos. These images then served as discussion triggers in interviews, deepening reflections on women’s roles and household dynamics.Social mapping, which engaged participants in identifying local actors (e.g., PKK groups, midwives, religious leaders) and community spaces that could support smoke-free home initiatives.

These participatory tools extended the study beyond descriptive accounts and encouraged women to articulate strategies, negotiate household norms, and envision actionable steps for creating smoke-free environments. Thus, the PAR approach in this study was not limited to documenting experiences but also aimed at stimulating critical reflection and collective awareness toward behavioral change.

### 2.2. Study Setting and Participants

This study was conducted in three districts in Aceh Province, Indonesia: Aceh Singkil, Gayo Lues, and Pidie Jaya. These areas were intentionally chosen to reflect a diversity of local cultural and socioeconomic conditions. A total of 75 female participants were recruited through purposive and snowball sampling techniques. The inclusion criteria included women aged 18–50 years who were married and residing in the same household as a family member who had quit smoking for at least one year, had lived in the selected location for a minimum of one year, were physically and mentally healthy, and were willing to participate and provide informed consent. This study is categorized as participatory because it engaged women in reflecting on their roles and eliciting their lived experiences and insights, which are expected to serve as the foundation for designing future promotive public health programs.

### 2.3. Data Collection and Saturation

This study collected data through in-depth, semi-structured interviews conducted between June 2023 and March 2024 in participants’ homes, using either Bahasa Indonesia or the Acehnese dialect, depending on each participant’s language preference. To enhance data credibility, each participant was interviewed twice in the first session, aiming to gather initial narratives, while the second was conducted to verify transcripts and confirm emerging themes. The smoking status of husbands was determined through subjective self-reports provided by their wives, based on daily observations within the household. No biometric assessments or clinical verification methods were employed, as this study’s qualitative orientation emphasized capturing the women’s lived experiences and perceived relational dynamics rather than objectively measured behaviors. In this context, smoking cessation was operationally defined as the husband’s conscious effort to refrain from smoking within the home, particularly in the presence of children or other family members. This definition reflects a behavioral and relational shift rather than clinically confirmed abstinence. In addition to interviews, the use of photovoice and social mapping provided participatory spaces for women to critically reflect on their roles and collaboratively identify feasible strategies for smoke-free households.

### 2.4. Data Analysis

Thematic analysis was conducted using NVivo 15 software to facilitate systematic data coding and the identification of themes. The process included first-cycle coding—the initial open coding of interview transcripts to identify relevant statements and meanings—and second-cycle coding, which involved grouping and refining codes into broader categories and themes. Themes were organized in alignment with Lawrence Green’s PRECEDE model, focusing on predisposing, enabling, and reinforcing factors that influence smoking cessation behaviors in the family.

### 2.5. Data Trustworthiness (Validity and Reliability)

To establish the validity and reliability of the qualitative data, Lincoln and Guba’s four trustworthiness criteria were applied: Credibility: Achieved through prolonged engagement, triangulation across participant backgrounds and locations, and member checking via follow-up interviews. Transferability: Enhanced by providing thick descriptions of cultural context and participant experiences from three diverse districts. Dependability: Ensured through a transparent audit trail of data collection and analysis procedures and the consistent application of the interview protocol. Confirmability: Supported by the use of direct participant quotations and reflective journaling to minimize researcher bias.

### 2.6. Ethical Considerations

This study received ethical approval from the Ethics Committee of Syiah Kuala University, No. 102/EA/FK/2023. All participants signed informed consent forms after being informed of the research objectives, procedures, and their right to withdraw at any time. Confidentiality was maintained by anonymizing all participant identifiers. This study adhered to the ethical principles outlined in the Declaration of Helsinki and its subsequent amendments.

## 3. Results

### 3.1. Subject Characteristics

This participatory qualitative study investigated the role of women in influencing smoking cessation within families in three underprivileged districts of Aceh Province—Aceh Singkil, Gayo Lues, and Pidie Jaya. Drawing on data from 75 female participants, the analysis revealed how women act as agents of change by leveraging their social roles to promote smoke-free households. The findings were categorized into three domains: (1) predisposing factors, including personal attitudes and knowledge about smoking; (2) enabling factors, such as access to information and household economic dynamics; and (3) reinforcing factors, including religious guidance and community norms. Women contributed significantly as caregivers, motivators, and informal health advocates, influenced by cultural, religious, and socioeconomic contexts. Supporting this, Table 1 presents the participants’ sociodemographic profiles, highlighting their maturity, diverse educational backgrounds, domestic responsibilities, and prior exposure to health education factors that shaped their capacity to support cessation behaviors in their households. These insights lay the groundwork for developing culturally grounded, community-based interventions in Aceh.

### 3.2. Data Validity

Table 2 presents the assessment of qualitative data trustworthiness based on Lincoln and Guba’s four criteria: credibility, transferability, dependability, and confirmability. Each aspect is clearly defined and accompanied by specific strategies implemented in this study to ensure methodological rigor in data collection and analysis processes. The table also outlines key findings linked to each criterion and illustrates how these were applied within the research context. Through strategies, such as prolonged engagement, triangulation, thick contextual descriptions, audit trails, and reflective journaling, this study ensures that the findings are both valid and reliable. This structured assessment strengthens the methodological integrity and supports the robustness of the participatory research outcomes in exploring women’s roles in controlling smoking behavior within Acehnese families.

### 3.3. Research Main Findings

Table 3 highlights five key themes that describe women’s strategies in promoting smoke-free households in Aceh. These include emotional motivations and assertive actions grounded in family health concerns, efforts to seek accurate health information, and the influence of religious teachings and cultural norms. Community support further legitimizes women’s advocacy, while personal empowerment enables them to become agents of change. Together, these factors illustrate a complex interplay of personal, social, and institutional elements that shape women’s leadership in tobacco control at the household level.

#### 3.3.1. Women’s Perceived Role as Health Advocates in the Family

Table 4 reports how women take on proactive roles as health advocates within their families through three distinct forms. First, as social control agents, they apply persuasive, emotional, and religious strategies to discourage smoking indoors, particularly to protect children. Second, they act as family health protectors by physically asserting boundaries, such as posting “no smoking” signs to create a safer, smoke-free home environment. Third, women serve as health education communicators, using media and storytelling to raise awareness among children and spouses and empowering children to reinforce anti-smoking messages. These roles reflect women’s growing agency and strategic efforts to safeguard family health.

The process of knowledge construction within households unfolded through ongoing dialogue and negotiation rather than unilateral imposition. Women often initiated discussions about smoking risks, which were then met with responses and adjustments from other family members. For example, some households negotiated specific smoke-free times (e.g., during meals) or agreed to relocate smoking outdoors, while others collaboratively created reminders such as handwritten signs. Many participants observed that this collaborative approach led to reductions in smoking frequency, gradual progress toward cessation, and the establishment of household smoke-free zones. Beyond these behavioral outcomes, the findings also highlight women’s growing sense of self-efficacy—witnessing tangible change reinforced their confidence to continue advocating for health within their families and communities.

Figure 2 illustrates the proportions of the most dominant roles played by women in supporting smoking cessation within their families. The chart reveals that 42% of the women acted as social control agents, emphasizing their responsibility to monitor and influence smoking behaviors among family members. Meanwhile, 33% served as family health protectors, focusing on maintaining a healthy, smoke-free environment at home. The remaining 25% took on the role of health education communicators, actively disseminating information and raising awareness about the dangers of smoking. This distribution reflects the multifaceted involvement of women in family health management, particularly in tobacco control. Their roles extend beyond traditional caregiving; they embody moral authority, provide behavioral guidance, and offer educational outreach. These findings highlight the strategic and culturally embedded role of women as health advocates within the household, especially in the Acehnese context.

#### 3.3.2. Influence of Predisposing Factors on Women’s Involvement

Table 5 outlines how predisposing factors influence women’s active participation in protecting their families from smoking exposure. The process begins with an awareness of the dangers of secondhand smoke, often gained through health education or community counseling. This newfound knowledge prompts a sense of personal concern, especially regarding the health of their children. As awareness deepens, women experience a stronger motivation to act, such as asking their husbands to avoid smoking around children. This leads to a growing sense of responsibility, where women perceive themselves as primary guardians of family health and feel morally obligated to intervene. These predisposing factors foster a transformation in women’s roles from passive household caretakers to proactive health advocates within the home.

Figure 3 illustrates the distribution of female participants based on their level of knowledge about the health risks associated with smoking. Among the 75 women interviewed, 35 demonstrated a high level of knowledge, 25 had moderate knowledge, and 15 showed a low level of knowledge. While this bar chart presents only a descriptive overview and does not imply a causal relationship, qualitative interviews revealed that participants with higher levels of knowledge tended to be more actively involved in supporting smoking cessation efforts within the household. This finding highlights the potential influence of health literacy on women’s roles as health advocates. The narratives suggest that improved understanding of smoking-related dangers empowers women to initiate persuasive communication, establish smoke-free zones, and engage family members in dialogue about the harms of tobacco. Therefore, while the chart does not statistically confirm causation, it visually supports the qualitative insight that enhanced awareness is often accompanied by increased motivation and engagement in promoting a healthier domestic environment.

#### 3.3.3. Supportive Enabling Conditions Facilitating Women’s Actions

Table 6 highlights that various enabling conditions significantly strengthen women’s involvement in promoting smoke-free households. Social support from family members and fellow women provides psychological encouragement to take action. Community-level campaigns and meetings enhance collective awareness and solidarity. Additionally, support from health institutions and the availability of educational materials help women translate knowledge into concrete actions. Environmental backing and moral messages from religious leaders further legitimize women’s roles as health advocates. Altogether, these factors create a supportive environment that empowers women to serve as active agents of change within their households.

#### 3.3.4. Reinforcing Support from Community and Religious Institutions

Table 7 reports reinforcing support from the community and religious institutions. Support from both the community and religious institutions plays a crucial role in strengthening women’s involvement in creating smoke-free households. At the community level, encouragement from village leaders and the integration of smoking-related issues into public health activities help establish strong social legitimacy for women to take action. Collective efforts, such as the installation of banners and public campaigns, further contribute to raising collective awareness. Meanwhile, religious institutions offer profound moral reinforcement. Messages from religious leaders emphasizing that smoking harms the family serve as powerful ethical justifications for women to advocate for change. Additionally, religious gatherings such as sermons and Friday prayers function as platforms for health advocacy, positioning mosques as centers for public education. This combination of social and spiritual reinforcement empowers women with greater confidence and motivation to take on their roles as guardians of family health.

#### 3.3.5. Perceived Impact on Smoking Behavior Within the Household

Table 8 highlights the tangible impact of women’s advocacy on smoking behaviors within households. Many participants observed a reduction in smoking frequency, often shifting the activity outdoors or away from children, while others successfully established smoke-free zones at home through rules and symbolic reminders. Several women reported that their husbands showed signs of cessation progress, with gradual reductions in daily cigarette use and, in some cases, achieving complete cessation. Beyond behavioral outcomes, the findings also underscore the perceived self-efficacy of women. Witnessing change reinforced their belief that their actions mattered, boosting confidence to continue advocating for health within their families and communities. Together, these outcomes illustrate both practical behavioral changes and the empowerment of women as effective agents in promoting smoke-free households.

The pie chart in Figure 4 presents the distribution of perceived changes in smoking behavior among household members, as reported by the participants. A total of 42.7% of women observed a reduction in smoking frequency, 29.3% reported complete cessation, and 28% created smoke-free zones within their homes. These findings demonstrate how women’s sustained advocacy efforts, through emotional appeals, persistent reminders, and health communication, lead to fundamental behavioral changes within the household. The highest proportion of responses indicating reduced smoking frequency suggests a gradual shift in family habits influenced by consistent interpersonal engagement. The observation of nearly a third of participants experiencing complete cessation highlights the effectiveness of culturally and religiously grounded persuasion. Meanwhile, the establishment of smoke-free areas in domestic settings demonstrates increased awareness and respect for the health of others, particularly children and elderly family members. Collectively, these outcomes underscore the pivotal role women play in fostering a healthier home environment and demonstrate their capacity to drive meaningful change through advocacy, communication, and the support of enabling social conditions.

#### 3.3.6. Challenges and Adaptive Strategies in Influencing Behavior

Table 9 reveals the barriers women encounter when advocating for smoke-free households, as well as the adaptive strategies they employ to overcome them. Many participants face gender-based resistance, where their efforts are dismissed due to patriarchal attitudes that minimize women’s authority. Cultural norms often frame smoking as a masculine right, making it difficult to challenge male behavior. Additionally, communication barriers, such as avoidance or dismissal during discussions, further complicate their efforts to advocate. Despite these challenges, women demonstrate resilience and creativity. They use emotional and child-centered appeals, highlighting the impact of smoke on their children’s health to evoke empathy. Others opt for indirect storytelling, using examples of neighbors or relatives to convey the dangers of smoking subtly. Some women strategically involve third-party influencers, such as religious leaders, to reinforce their message with social and moral authority. These adaptive approaches illustrate women’s agency in navigating complex household dynamics to protect their families’ health.

#### 3.3.7. Transformation of Gender Role Perception

Table 10 highlights a significant shift in women’s perceptions of their roles within the household, particularly regarding smoking behavior control. Women have begun to exhibit greater self-confidence and awareness of their personal agency (Emerging Empowerment), challenge longstanding patriarchal norms that have previously silenced their voices (Challenging Traditional Roles), and expand their roles as protectors of family health and role models for other women (Role Expansion). This transformation reflects a movement toward more active female engagement as change agents in fostering a healthier and more equitable home environment.

Interview results revealed varying levels of smoking behavior change among husbands. Most participants reported a reduction in smoking frequency or relocation of smoking activities to outdoor areas. At the same time, a smaller number indicated that their husbands had entirely quit smoking, ranging from 6 months to over a year. These findings highlight a continuum of behavioral change rather than a uniform cessation pattern.

Figure 5 illustrates the evolving perception of gender roles among women participants, particularly in relation to their involvement in family health decisions. A majority of respondents (60%) reported increased confidence in their role as health advocates, signaling a significant shift in self-perception and agency within their households. These women now view themselves not only as caregivers but also as key actors in shaping family health behaviors, especially in promoting smoke-free environments. Meanwhile, 25% of participants expressed that they now share an equal role in health-related decision making with their husbands, indicating progress toward gender parity in household leadership. This suggests that health advocacy can be a powerful entry point for negotiating shared authority in domestic matters. Only 15% of respondents noted no significant change in their perceived roles, which may reflect persistent barriers such as strong patriarchal norms, lack of support, or limited exposure to health education.

## 4. Discussion

The discussion in this study centers on the transformative role of women as key agents in promoting tobacco control within the household setting, drawing on qualitative insights from Aceh, Indonesia. In a region deeply rooted in cultural and religious values, women were found to actively shape smoke-free environments through advocacy, emotional communication, and social persuasion. Their actions reflect a dynamic interplay between gender, health literacy, and community norms, where empowerment serves as both a catalyst and a product of health interventions at the family level. By examining how women negotiate influence within traditionally male-dominated households, this study situates their contributions within broader efforts to promote public health equity and behavioral change in low-resource, culturally embedded contexts.

Notably, the smoking status of husbands in this study was reported through the narratives of their wives, based on everyday observations within the household. No objective verification methods, such as clinical assessments or direct observation, were employed. This approach is consistent with this study’s qualitative aim to capture lived experiences and perceived realities, rather than to quantify smoking behavior. Rather than a limitation, the subjective nature of these data reflects the central focus on understanding how women interpret, respond to, and act upon smoking within their domestic environments.

Building on this, this study explicitly highlights the role of women as agents of change in household-level tobacco control. Interview findings revealed that women were not merely passive recipients of secondhand smoke exposure but actively contributed to shaping smoke-free home environments through emotional communication, social persuasion, and health advocacy. These narratives highlight the importance of empowering women in household health decision making and demonstrate how agency is exercised through relational strategies and informal negotiation, thereby underscoring their central role in promoting healthier family practices.

Women play a central role as agents of change in promoting family health, particularly in reducing exposure to tobacco smoke within the household. Through emotional appeals, interpersonal communication, and role modeling, they actively educate family members and establish rules to create a smoke-free home environment. Previous studies have shown that women often serve as key drivers of health promotion due to their moral and emotional roles as mothers and wives [33]. Therefore, empowering women through enhanced health literacy and social support is crucial for strengthening their impact on behavioral change toward a healthier family lifestyle.

Knowledge and attitudes serve as key predisposing factors influencing women’s participation in advocating for smoke-free environments. The higher their understanding of the health risks associated with smoking, particularly secondhand smoke, the more proactive and confident they become in initiating behavioral change within the household. This correlation is supported by previous research, which highlights that increased health literacy significantly enhances women’s engagement in health-promoting behaviors [34]. Thus, educational interventions that improve women’s awareness are essential to fostering their role as effective advocates for smoking cessation in family settings.

Enabling conditions such as access to health information, prior involvement in education campaigns, and exposure to community outreach programs significantly enhance women’s capacity and motivation to advocate for smoke-free households. These factors not only equip women with the necessary knowledge but also build their confidence and communication skills, enabling them to influence family behavior. Studies have shown that environments with supportive health infrastructure and regular public health messaging empower women to take active roles in shaping household norms and protecting family well-being [35]. Such conditions are, therefore, essential in translating awareness into concrete advocacy actions.

Social and religious legitimacy plays a crucial role in strengthening women’s ability to take assertive action in promoting health within the family, particularly regarding smoking behavior. Support from spiritual leaders, customary laws, and prevailing community norms provides women with moral and social justification to speak out and influence change. This legitimacy not only protects them from social resistance but also amplifies their authority in traditionally patriarchal contexts. Prior research emphasizes that religious and cultural endorsement can significantly boost women’s participation in public health advocacy, turning personal concern into socially recognized action [36].

Women in this study reported tangible changes in household smoking behaviors as a result of their advocacy efforts. These changes included a reduction in smoking frequency, the establishment of designated smoke-free zones, and, in some cases, complete cessation, particularly among their spouses. Such transformations were attributed to a combination of persistent communication, emotional appeals, health education, and social reinforcement. These findings underscore the influential role women play in shaping healthier domestic environments, aligning with previous studies that highlight the effectiveness of female-led health interventions within families [37].

Women employed adaptive strategies to overcome social and patriarchal barriers when promoting smoke-free behaviors within the household. Faced with resistance from their husbands or constrained by traditional gender norms, they resorted to indirect yet effective methods, such as initiating informal discussions, expressing emotional concerns related to family health, and involving influential third parties, like religious leaders or children. These approaches reflect their ability to navigate complex power dynamics while maintaining cultural harmony. Previous studies have similarly noted that women often rely on relational communication and community figures to assert influence in health matters within patriarchal settings [38].

Women’s involvement in health advocacy represents a significant transformation in gender roles, shifting from traditional domestic responsibilities to emerging leadership roles in the family and community health domain. This transition is characterized by increased confidence, assertiveness, and influence in shaping health-related decisions, particularly regarding smoking behaviors. Rather than being confined to caregiving tasks, women are now seen and see themselves as proactive health agents. This role expansion aligns with broader patterns of gender empowerment observed in the health promotion literature, where women’s participation in public health efforts is linked to both individual agency and structural support [39]. Such a shift not only enhances family well-being but also contributes to long-term gender equity in decision-making spheres.

The role of emotion and interpersonal communication emerges as a crucial strategy in women’s advocacy for healthier family environments. By employing messages grounded in affection, empathy, and concern, particularly for children’s well-being, women can persuade family members to reduce or quit smoking effectively. Emotional appeals resonate more deeply within familial relationships, fostering a sense of guilt or responsibility in smokers. For instance, reminders framed around the health risks to children or expressions of maternal concern tend to generate more receptive responses. These communication styles, rooted in empathy and relational bonds, have been identified in previous studies as practical tools for initiating behavior change within domestic settings [40]. Thus, emotional and empathetic communication strengthens women’s influence and reinforces their role as trusted advocates in the household.

Community-based and gender-sensitive health education programs have shown a significant impact in empowering women as key agents of change in promoting healthy behaviors within families. When women are actively involved as participants, facilitators, or peer educators, these programs become more relatable and culturally aligned, increasing their acceptance and sustainability. Research has demonstrated that such gender-inclusive approaches enhance the credibility of health messages and improve adherence to behavior change, especially in patriarchal communities where women’s voices may otherwise be marginalized [41]. By integrating women’s lived experiences and leadership roles into health interventions, these programs not only address immediate issues, such as tobacco exposure, but also contribute to long-term shifts in gender norms and family health decision-making dynamics.

This study demonstrates the transformational dimension of Participatory Action Research (PAR) in several ways. First, the use of photovoice and reflective discussions allowed women to critically examine their household environments and articulate smoking-related risks that were previously taken for granted. Second, through iterative dialogues, women negotiated concrete behavioral changes within their families, such as establishing smoke-free zones inside the home, limiting smoking to outdoor areas, and introducing household reminders (e.g., notes on the wall). Third, the process of social mapping expanded women’s awareness of potential community-level support systems, including PKK groups, midwives, and religious leaders, thereby linking individual household practices to wider social structures. These participatory processes go beyond documenting lived experiences; they stimulated critical reflection, fostered women’s agency in health decision making, and catalyzed collective awareness of the importance of creating smoke-free environments. Thus, this study embodies the transformational goal of PAR by translating participants’ reflections into tangible shifts in household norms and social consciousness.

This study contributes to the development of gender and health literacy in local communities by emphasizing the importance of culturally grounded, gender-responsive approaches in public health interventions. By recognizing women not merely as passive recipients but as active health advocates shaped by local norms and values, the research highlights how gender-sensitive frameworks rooted in community identity can enhance the relevance and effectiveness of health programs. The integration of local wisdom, religious teachings, and familial structures into advocacy strategies fosters trust and engagement, ultimately leading to more sustainable behavioral changes [42]. These findings align with global recommendations that stress the intersection of gender equity and health education in achieving community health resilience [43].

## 5. Conclusions

Based on the findings, this study concludes that women play a central role as agents of change in controlling smoking behavior at the household level in Aceh, Indonesia. They actively take on roles as social regulators, health communicators, and family protectors through empathetic interpersonal approaches, knowledge of smoking hazards, and the social and religious legitimacy they receive. Predisposing factors such as awareness levels; enabling conditions such as access to health information and prior educational experiences; and support from social institutions, including religious leaders and community norms, further strengthen their capacity to act.

Tangible behavioral changes, such as reduced smoking frequency, the establishment of smoke-free zones, and even complete cessation among spouses, demonstrate the effectiveness of women’s advocacy efforts. Despite facing social barriers and patriarchal norms, women developed adaptive strategies, including informal discussions, emotional appeals, and the involvement of third parties (e.g., religious leaders or children). This also reflects a transformation in gender roles from traditional domestic roles to leadership in family health matters. This study highlights that empowering women through health literacy and community support significantly contributes to creating smoke-free home environments and fostering healthier family behavior.

## Figures and Tables

**Figure 1 ijerph-22-01490-f001:**
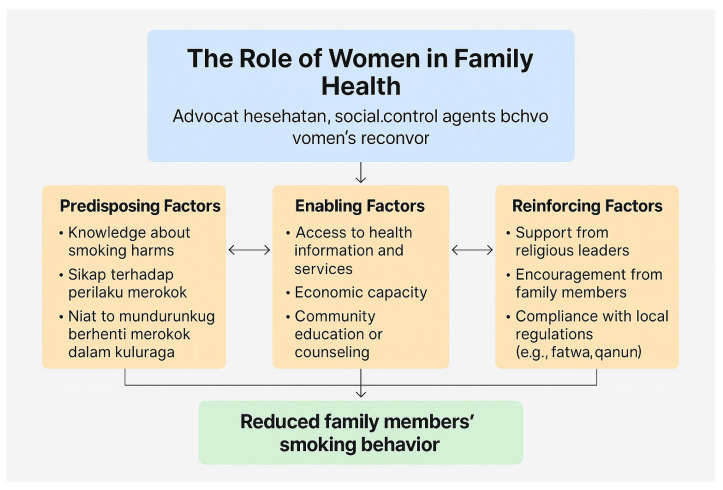
A conceptual framework of the role of women in family health, encompassing predisposing, enabling, and reinforcing factors that contribute to the reduction of smoking behavior within the household environment.

**Figure 2 ijerph-22-01490-f002:**
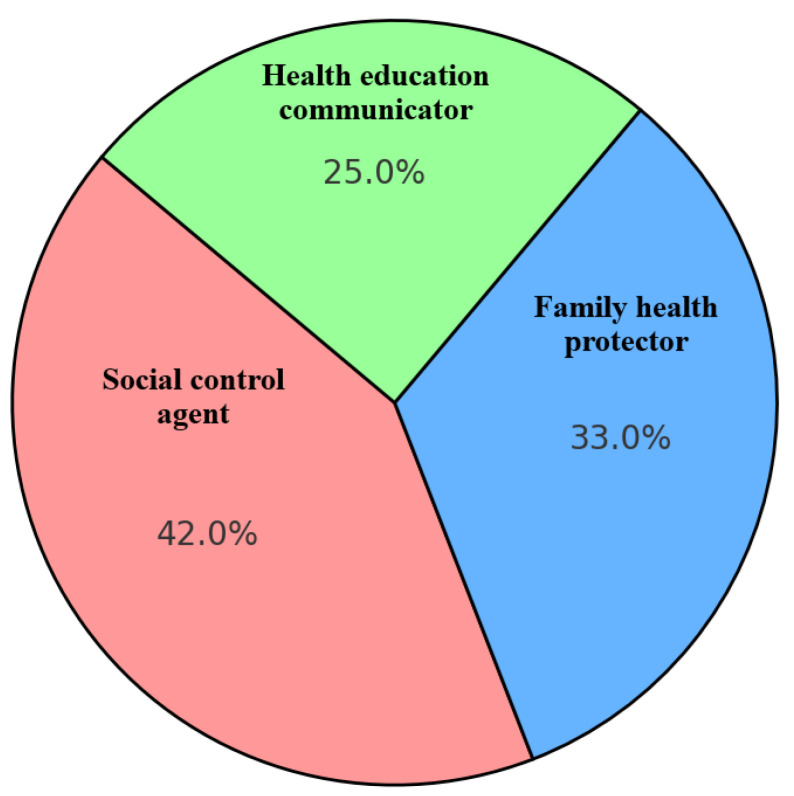
Proportions of dominant roles played by women in supporting smoking cessation in the family.

**Figure 3 ijerph-22-01490-f003:**
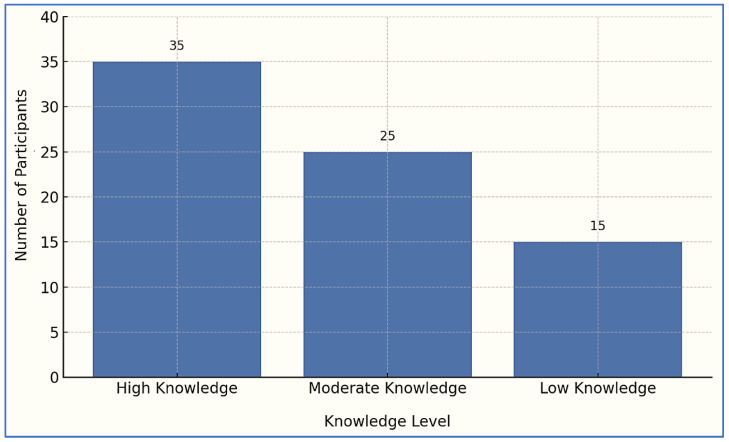
Frequency of participants by knowledge level on smoking risks.

**Figure 4 ijerph-22-01490-f004:**
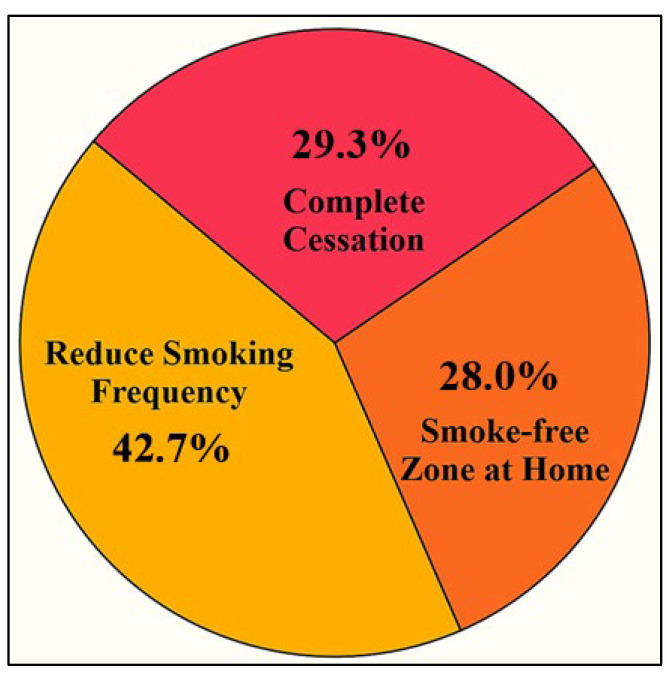
The proportion of reported smoking behavior changes within the household.

**Figure 5 ijerph-22-01490-f005:**
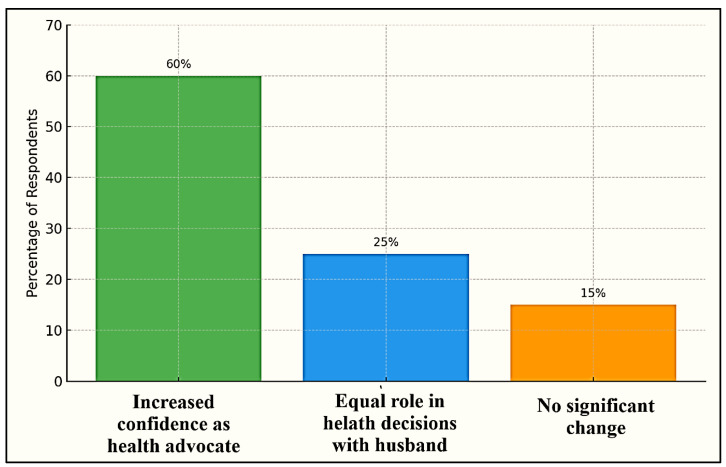
Transformation of gender role perception among women participants.

**Table 1 ijerph-22-01490-t001:** Characteristics of study participants.

Variable	Category	N	Percentage (%)	Reason Required
District	Aceh Singkil	20	26.7	Represents diverse geographical and cultural settings in the Aceh Province
	Gayo Lues	30	40.0	
	Pidie Jaya	25	33.3	
Age Group	18–29 years	21	28.0	Age influences maturity, experience, and responsibilities within the household.
	30–39 years	27	36.0	
	40–50 years	27	36.0	
Education Level	Primary school	25	33.3	Educational attainment affects health literacy, attitudes, and communication capacity.
	Secondary school	34	45.3	
	Higher education	16	21.3	
Employment Status	Housewife	33	44.0	Employment status reflects both economic capacity and social role, influencing household behavior.
	Informal worker	26	34.7	
	Civil servant/teacher/etc.	16	21.3	
Exposure to Health Education	Yes	49	65.3	Assesses the impact of public health campaigns or counseling on behavior awareness
	No	26	34.7	
Years Living in Community	<5 years	15	20.0	The duration of residence influences community ties, access to information, and cultural adaptation.
	5–10 years	30	40.0	
	>10 years	30	40.0	
Religious Involvement	Active	38	50.7	Religious involvement affects value orientation, including perceptions of smoking.
	Occasionally active	26	34.7	
	Not Active	11	14.6	
Community Participation	Yes	49	65.3	Participation in social groups (e.g., health posts and religious gatherings) supports social control roles.
	No	26	34.7	
Family Smoker	Husband/former smoker	75	100.0	Central to the study: assessing women’s role in supporting smoking cessation in family settings

Data Processing, 2025.

**Table 2 ijerph-22-01490-t002:** Trustworthiness of the qualitative data.

Trustworthiness Aspect	Definition	Process	Findings	Application in This Study
Credibility	Confidence in the truth and authenticity of the data and research findings.	- In-depth interviews conducted twice (verification & clarification) - Triangulation across locations and participant backgrounds - Member checking of interview results	Consistent data and emerging thematic patterns across participants from different regions.	Achieved through prolonged engagement, regional triangulation, and participant data verification.
Transferability	The extent to which findings can be applied to other similar contexts.	- Selection of regions with socioeconomic variation - Detailed description of local context - Comprehensive profiling of participant backgrounds	The findings may apply to other areas with similar social conditions.	Ensured through rich contextual descriptions of three underprivileged districts in Aceh.
Dependability	Consistency of the research process and results over time.	- Documentation of procedures - Use of a consistent interview protocol - Systematic storage of recordings and transcripts	Findings are traceable to original data and show analysis stability.	Maintained through audit trails and consistent interview procedures across all research subjects.
Confirmability	The neutrality of the data: ensuring findings originate from participants, not researcher bias.	- Reflective journaling - Direct quotations from participants - Cross-validation of codes and themes by research team	The findings reflect the participants’ voices and perspectives without the researcher’s influence.	Upheld through researcher reflexivity, documentation, and the use of direct participant quotations.

Data Processing, 2025.

**Table 3 ijerph-22-01490-t003:** Summary of main themes, sub-themes, and representative quotes from participants.

Main Themes	Sub-Themes	Descriptions	Illustrative Quotes
Women’s Role in Household Tobacco Control	Emotional Motivation	Concern about husband’s and children’s health; fear of disease	“I told him, if you love your kids, stop smoking.” (P-13, Aceh Singkil)
	Domestic Negotiation	Dialogues, compromises, or conditions to restrict smoking	“I said, you can smoke outside, not in front of the kids.” (P-27, Pidie Jaya)
	Assertive Persuasion	Open confrontation or assertive language	“I was angry when he smoked near our baby. I threw the cigarette.” (P-45, Gayo Lues)
Health Literacy and Information Seeking	Seeking Trusted Sources	Use of health workers, religious leaders, or TV programs	“I heard from the midwife that secondhand smoke is dangerous.” (P-31, Gayo Lues)
	Misinformation or Lack of Access	Lack of proper knowledge or exposure to misleading info	“They say herbal cigarettes are safe, so he switched to those.” (P-8, Aceh Singkil)
Cultural and Religious Framing	Religious Advice	Using Islamic teachings to discourage smoking	“I reminded him that harming health is against Islam.” (P-22, Pidie Jaya)
	Gender Norms	Reinforcement of masculine decision-making	“It’s hard. He is the man of the house.” (P-60, Aceh Singkil)
Community-Level Influence	Peer Pressure and Role Models	Observing neighbors/friends who quit smoking as examples	“He stopped because his brother stopped first.” (P-48, Gayo Lues)
	Community Support and Sanctions	Community norms, bans, or support groups	“In our village, the mosque banned smoking near the entrance.” (P-19, Pidie Jaya)
Participatory Empowerment	Reflective Learning	Women reflect on their own power and voice	“I never thought I could make him quit. But I did.” (P-7, Gayo Lues)
	Aspirations for Advocacy	Hope to help others and become health agents	“I want to help other mothers do the same.” (P-33, Aceh Singkil)

Data Processing, 2025.

**Table 4 ijerph-22-01490-t004:** Women’s role as health advocates in the family.

Role	Description	Quote
Social Control Agent	Women implement unwritten rules to limit smoking indoors, especially near children, using persuasive, emotional, and religious-based approaches.	“If my husband wants to smoke, I tell him not to do it inside the house. I feel sorry for the children...” (P34), Aceh Singkil.
Family Health Protector	Women proactively create a smoke-free home environment by posting homemade signs, signaling their health priorities, and protecting their children.	“I put a paper sign saying ‘no smoking’ near the kitchen... our children’s health matters more.” (P11), Pidie Jaya
Health Education Communicator	Women use media (videos, stories, experiences) to educate both children and spouses about the harms of smoking, empowering children to remind fathers.	“I once had the children watch a video... So they’d be afraid and remind their father too.” (P07), Gayo Lues.

Data Processing, 2025.

**Table 5 ijerph-22-01490-t005:** Influence of predisposing factors on women’s involvement.

Theme	Sub-Theme	Meaning/Insight	Quote
Influence of Predisposing Factors	Awareness of smoking risks	Knowledge gained from health education or community-based counseling triggers concern.	“I didn’t know how harmful secondhand smoke was until I heard it from the clinic.”—P05
	Personal health concern	Women begin to relate the risks of smoking to real dangers affecting their children.	“Now that I understand the risks, I feel I must protect my children.”—P19.
	Motivation to take action	Internal drive increases after understanding the consequences of smoking in the household	“After learning about the effects, I told my husband to stop smoking around the kids.”—P27
	Perception of responsibility	Women perceive themselves as protectors of family health and feel compelled to act	“If not me, who else will warn him? I’m responsible for our children’s safety.”—P33

Data Processing, 2025.

**Table 6 ijerph-22-01490-t006:** Supportive enabling conditions facilitating women’s actions.

Theme	Sub-Theme	Quote
Social Support	Support from family members	“My eldest child told his father not to smoke inside, that really helped.”—P09.
	Encouragement from other women	“My neighbor also told her husband to stop smoking, so I followed her steps.”—P15.
Community Support	Village leader’s campaign against smoking	“Our village head told us to create a smoke-free home.”—P12
	Community meetings raising awareness.	“We often discuss this in village women’s gatherings.”—P17.
Institutional Encouragement	Support from local health institutions	“The health clinic staff encouraged me to talk to my husband.”—P20
Access to Educational Resources	Availability of visual aids or pamphlets	“We got leaflets and stickers from the community meeting.”—P11
Environmental Support	Clean air campaigns in the neighborhood	“The whole neighborhood is now trying to keep the air clean.”—P29
Faith-based Encouragement	Religious messages supporting health	“Our imam once said that smoking harms the family, which touched many hearts.”—P18

Data Processing, 2025.

**Table 7 ijerph-22-01490-t007:** Reinforcing support from community and religious institutions.

Theme	Sub-Theme	Quote
Community Reinforcement	Public endorsement of anti-smoking behavior	“When the village chief supports no-smoking rules, everyone pays more attention.”—P10.
	Integration in community health activities	“We included smoking discussions during health events in the village.”—P23.
	Visibility of collective actions	“We put up banners about smoke-free homes around the neighborhood.”—P06.
Religious Reinforcement	Religious leaders promoting health values	“Our imam said smoking is harmful, and as Muslims we should avoid harming others.”—P16.
	Alignment with religious morals	“It’s easier to advise my husband when I remind him of what the ustadz said at the mosque.”—P28.
	Religious gatherings as platforms for advocacy	“We often hear health messages during Friday prayers.”—P30.

Data Processing, 2025.

**Table 8 ijerph-22-01490-t008:** Perceived impact on smoking behavior.

Theme	Sub-Theme	Quote
Behavioral Change in Household	Reduced frequency of smoking	“He still smokes, but now only outside the house—not around the children anymore.”—P12
	Smoke-free zones created at home	“I put up a no-smoking sign in our living room, and he respects it now.”—P28.
Cessation Progress	Gradual reduction leading to quitting	“He used to smoke five times a day, now maybe once, sometimes not at all.”—P39.
	Complete cessation achieved	“He hasn’t touched a cigarette in six months since I talked to him seriously.”—P41.
Perceived Self-Efficacy of Women	The belief that action makes a difference	“I didn’t think he’d listen, but now I believe I can influence change.”—P07.
	Increased confidence in advocacy role	“Seeing him stop gave me the confidence to keep reminding others too.”—P24

Data Processing, 2025.

**Table 9 ijerph-22-01490-t009:** Challenges and adaptive strategies in influencing smoking behavior within the household.

Theme	Sub-Theme	Quote
Challenges in Advocacy	Gender-based resistance	“He told me I’m just a woman, I shouldn’t be telling him what to do.”—P34.
	Cultural and patriarchal norms	“In our culture, men think smoking is their left. It’s hard to change that.”—P22.
	Communication barriers	“He just walks away when I bring up the topic of smoking.”—P18.
Adaptive Strategies	Emotional and child-centered appeal	“I waited until he was calm. Then I told him how our child coughs whenever he smokes inside.”—P41
	Indirect discussion and storytelling	“I didn’t argue. I shared a story about a neighbor’s husband who got sick from smoking.”—P07.
	Involving third-party influencers	“When the imam mentioned it during the sermon, he started to feel embarrassed.”—P53.

Data Processing, 2025.

**Table 10 ijerph-22-01490-t010:** Transformation of gender role perception.

Theme	Sub-Theme	Quote
Emerging Empowerment	Increased confidence in decision-making	“I used just to follow what he said, but now I speak up for our children.”—P25.
	Recognition of personal agency	“I realized I could change things at home, even if I’m just a housewife.”—P14.
Challenging Traditional Roles	Questioning gendered expectations	“Why is it only men who can decide everything? We care about health too.”—P38.
	Breaking the silence on harmful habits	“Before, I stayed quiet. Now, I tell him directly that smoking is dangerous.”—P47.
Role Expansion	Taking leadership in household health	“I’m the one who reminds everyone about health now, including my husband.”—P30.
	Becoming a role model for other women	“I hope my story encourages more women to speak up.”—P12

Data Processing, 2025.

## Data Availability

The data supporting this study are not publicly available due to ethical restrictions.

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
