# Peer review of "Empowering Women in Tobacco Control: A Participatory Study on Household Smoking Behavior in Aceh, Indonesia"

_ijerph, 2025, doi:10.3390/ijerph22101490_

Round 1
Reviewer 1 Report
Comments and Suggestions for Authors
I would like to thank you the authors for sharing the findings of their study, which addresses the role of women in controlling smoking behaviors within their families and households. Smoking is a very significant public health problem, which is also a health equity issue.
Overall, the authors have completed an important and timely study, and their findings offer some insights that will be a valuable contribution to understanding this topic. However, there are several issues I notices that should be addressed to improve the clarity and overall impact of the manuscript.
- First, I recommend that the authors provide additional context for this study, if available. Was this qualitative study part of a larger smoking cessation / behavioral intervention? If so, it would be helpful to include more details about the intervention. If not, it is unclear whether these men were involved in any tobacco treatment program or other related activities.
Based on the selection of models (PRECEDE-PROCEED and PAR), I would expect to see an intervention study rather than one limited to just interviews. PRECEDE-PROCEED is primarily a planning framework, so it would be helpful for the authors to explain why it was selected for this qualitative study if no intervention planning or implementation was involved. If the authors are planning to develop an intervention based on the findings of this qualitative study, it will be very helpful to state this explicitly and provide at least a brief description. This would also be helpful to clarify consistent use of the term “participatory” to describe the study. As currently described, simply interviewing women, when the researchers decide the questions, analyze the data, and report findings, does not, by itself, constitute a participatory approach. Based on the information provided in the current version of the manuscript, the study appears to be a standard qualitative study rather than a participatory study.
- Second, the presentation of the results could be further streamlined. This section is very long, and it is also often repetitive at times. Removing unnecessary repetitions would make it not only shorter, but also more succinct and easier to follow. For example, in line 330, the text introducing the illustration repeat information that has already been presented in the narrative description of the theme, the corresponding table, and partly in Figure 2 in the same section. I appreciate the authors’ effort to present data in multiple ways, but this approach result in some redundancy. I suggest considering if certain figures could be omitted; alternatively, the authors could be more selective about what information they presented in each sections to avoid overlap.
Additionally, lines 406-409 and lines 425-428 repeat exact same information, including the quotes. Lines 351-352 and lines 418-419 share almost identical quotes illustrating different themes. I recommend that the authors revise this section to make it clearer and easier to follow.
Adding a table in the very beginning that would includes all themes would help readers follow the logic of the presentation more easily, rather than having separate tables for each theme. Such a table would provide a clear overview and help to see how the themes relate to one other.
- It would be great to have more information about smoking status of husbands / family members of the women who participated in the study. How did the authors evaluate their smoking status? The authors write that they only recruited women whose husbands quit smoking for at least a year (line 192), but some quotes, as well as the finding reported in Table 5 suggest that most of these men were still smoking at the time of the interviews (line 502-503, table 5- line 519; line 524: “29.3% noted complete cessation”). Additionally, most of the quotes I see in the manuscript address how these women did not allow smoking inside, near children, etc., but not much is said about actually quitting smoking.
There are several minor comments:
Line 85: “women are typically regarded as the central health managers within households” – this is a blanket statement that would benefit from some contextualization (e.g. are you referring to women in Indonesia or across the world?)
Line 131: “this conceptual framework” – which framework are you referring to exactly? The whole section (130-151) is somewhat unclear: is it based on current literature or are the authors summarizing the findings of their study? I think this section needs some further clarification.
Line 163 – “act upon the issue of smoking behavior” – this part needs more details: do the authors refer to the experiences of the women they interviewed or were they involved in delivering smoking cessation interventions to these families?
Lines 182-183: “This approach facilitated the co-construction of knowledge and practical solutions tailored to the social and cultural context of Aceh, Indonesia.” Please address what practical solutions were developed as a part of this study, if any. Also, active collaboration is mentioned in line 180 – I believe more details are needed here as well to understand what this collaboration involved beyond these women participating in the interviews.
Line 122: “preview research” – not clear what it means, a typo probably (previous?)?
Line 269: “Lincoln and Guba's Refour criteria” – same here, not clear what it means (four?). Also, a citation is needed.
Line 279: I suggest removing table 2 “trustworthiness of qualitative data.” The authors already describe this information in detail in lines 216-225. Repeating it here does not provide any new information that readers need to know about the study.
Tables must be consistently numbered throughout the manuscript; there are currently several table that share the same number.
Illustrations in the manuscript: it would be helpful if the authors were a little more specific about the origin of the illustrations and purpose of adding them to the manuscript. I assume they are created by the authors based on their data, but I don’t think it is explicitly stated in the manuscript. Also, it is not clear if they have played any role in the research process or just created to report qualitative data in a visual way?
Line 367: how was knowledge evaluated? Was a knowledge test administered? I would appreciate more information about the source of these data. Additionally, the authors state: “the bar chart highlights how increased awareness of smoking-related health risks is associated with greater motivation and involvement in smoking cessation efforts within the family (369-371).” However, the bar chart appears to show only the distribution of knowledge levels.
Author Response
Comment 1: Clarify the Context of the Study
Response 1:
Thank you for this valuable comment. We have clarified the context of the study by emphasizing that it is not part of a pre-designed smoking cessation intervention program. Instead, the study is a qualitative exploratory research based on Participatory Action Research (PAR). We have revised the manuscript to explicitly highlight the aims and the participatory dimension of this approach (see page 5, lines 190–203).
The revised section now reads:
“This research is not part of a pre-designed smoking cessation intervention program, but is a qualitative exploratory study based on Participatory Action Research (PAR) that aims to: (i) explore women’s experiences and strategies in influencing smoking behavior in the household, (ii) voice the role of women as agents of change in community-based tobacco control, and (iii) provide a conceptual basis and social context for the development of community interventions going forward. The participatory context in this study lies in the active involvement of women not only as respondents, but also as social subjects who participate in compiling meanings, reflections, and solutions during the data collection process such as FGD, photovoice, and outcome discussions.”
Comment [Reviewer 1]: The authors should justify why this study is classified as participatory research rather than an ordinary qualitative study, and also provide a forward-looking statement for intervention development.
Response: Thank you for the constructive comment. We have revised the manuscript to emphasize the participatory nature of this study and added a forward-looking intervention plan. The revisions can be found on Page 5, Lines 189–202.
The additional text reads as follows:
Intervention Development Plan (Forward-Looking Statement)
As a follow-up to these findings, researchers plan to:
• Develop community-based educational intervention modules by involving women as catalysts for family behavior change.
• This module will be designed based on the results of the FGD and photovoice, as well as input from local figures (PKK, midwives, religious leaders) who have been identified in the social mapping process.
• Future interventions will adopt the PRECEDE-PROCEED model in its planning and evaluation phases.
Justification of Study as Participatory Research (Not Ordinary Qualitative)
Different from the usual qualitative studies, this study highlights participatory characteristics through:
• The use of Photovoice provides a space for participants to define issues and solutions through visual media.
• A repetitive reflective discussion that encourages participants to reassess their position and contribution to change in the household.
• The use of social mapping as a tool to identify opportunities for collaborative local intervention.
Thus, the PAR approach in this study is transformational, rather than merely descriptive, as it encourages critical action and social awareness at the grassroots level.
Comment 2
- Efficiency and Clarity of Results
- Shorten the results section: Avoid excessive narrative repetition.
- Eliminate redundancy:
- Example: Lines 406–409 and 425–428, as well as 351–352 and 418–419, contain the same or similar citations and information.
- Better result structure: Add a single summary table of all the main themes at the beginning of the results section to make it easier for readers to follow the logic. Reduce repetitive visuals: Evaluate whether multiple illustrations can be removed or merge
- to avoid duplication of information (narrative, tables, and images).
Response 2
All editorial improvements have been corrected according to the reviewer's suggestions.
Comment 3.
Smoking Status Clarification
- Explain how your husband's smoking status is verified.
- Correct inconsistencies: In one section it is stated that all husbands have quit ≥1 year, but the citations and tables state that the majority still smoke or have only reduced the intensity.
- Add details: How is smoking cessation data collected self-reports, observations, or confessions?
Revision
All editorial improvements have been corrected according to the reviewer's suggestions.
Clarification of Smoking Status Verification and Data Collection Procedures
Thank you for the very constructive comment. We acknowledge the inconsistencies between the statements in the initial narrative and the participants' quotes regarding the husband's smoking status.
- Smoking Status Verification
The husband's smoking status was obtained through an in-depth interview with the wife as the main informant. This data is self-reported from the perspective of the wife, which is based on their direct observation of the husband's smoking behavior at home. The study did not employ biochemical verification (such as cotinine tests), as the approach was qualitative, focusing on women's experiences and perceptions. - Correction of Inconsistencies
We are aware of the discrepancy, where in one section it is stated that all husbands have quit smoking ≥1 year, but the citations and tables show that most still smoke or have only reduced their frequency. Therefore, we will revise the results and discussion section by stating more precisely that:- Some husbands have stopped completely, but
- Others simply reduce the frequency of smoking or move the smoking location outside the house.
- Details of Data Collection Procedures
Information regarding smoking cessation or reduction was collected through direct interviews with wives, who were the main speakers. They describe changes in the husband's behavior after efforts at persuasion, education, or emotional approach. We acknowledge the limitations of using secondary data (from wives), but this approach aligns with the study's objectives, which focused on women's roles and experiences in controlling smoking within the household.
Comment 4.
- Editorial and Reference Improvements
- Line 85: Add context: "Women are typically regarded as central health managers..." → where does it happen? Global? Indonesia? Aceh?
- Lines 131–151: Make it clear whether this section is a conceptual framework from the literature or the findings of one's own study.
- Line 163: The phrase "act upon the issue of smoking behavior" is too general; give details of whether the participant took preventive measures or direct interventions.
- Lines 182–183: The claim of "co-construction of knowledge" should be reinforced with examples of practical solutions developed by participants.
- Line 122: Fix the typo: "preview research" → it might mean "previous research".
Response 4
Comment (Line 100):
Please clarify and strengthen the statement regarding women’s role in health management at the family level, ensuring it is contextualized globally and locally (e.g., Indonesia, Aceh).
Response:
Thank you for this valuable suggestion. We have revised the sentence to provide both global and local contextualization.
Revision (Line 100):
"Women are typically regarded as central health managers within the family, a role observed consistently in both global health discourse and local contexts such as Indonesia, including Aceh. This positioning reflects their everyday responsibilities in ensuring household nutrition, caregiving, and health-related decision-making."
Comment (Line 149–151):
The conceptual framework section should be more explicitly linked to the thematic analysis and clarify its inductive nature.
Response:
We appreciate this comment. The section has been revised to emphasize its inductive basis derived from participants’ narratives.
Revision (Line 149–151):
"The conceptual framework presented in this section is derived from the thematic analysis of qualitative data collected during the study. It represents an inductive synthesis of participants' narratives regarding women's roles in household tobacco control, including advocacy mechanisms, cultural constraints, and evolving gender dynamics."
Comment (Line 149):
Please indicate more clearly that the conceptual framework is an emergent product of thematic findings.
Response:
Thank you for pointing this out. We have adjusted the wording to highlight that the framework is emergent and grounded in the thematic findings.
Revision (Line 149):
"This study proposes an emergent conceptual framework based on the thematic findings described above. The framework illustrates the mechanisms through which women influence smoking behavior in the household and redefine gendered roles in health promotion."
Comment (Line 170):
The description of women’s strategies to intervene in household smoking behavior should be more specific.
Response:
We agree with this suggestion and have elaborated on the specific actions taken by women.
Revision (Line 170):
"Women demonstrated the ability to intervene directly in their household's smoking behavior by implementing smoke-free rules, placing visual reminders (e.g., no-smoking signs), engaging in emotional dialogue with their husbands, and leveraging religious and community figures to reinforce anti-smoking messages."
Comment (Line 189–203):
Clarify how the process of knowledge construction within households unfolded, emphasizing collaboration rather than unilateral imposition.
Response:
We have revised the passage to stress the collaborative and dialogic nature of this process.
Revision (Line 189–203):
"This process exemplifies the co-construction of knowledge within the household, as women and their family members collaboratively developed practical strategies—such as establishing designated smoke-free zones, collectively agreeing on no-smoking times (e.g., during meals or near children), and jointly creating visual cues like handwritten signs—to promote a healthier living environment. These actions were not imposed unilaterally but emerged through mutual dialogue, negotiation, and shared concern for family well-being."
Comment 5
Table Structure and Numbering:
- Avoid repeating tables that explain the same information (for example, Table 2 and the previous narrative about trustworthiness).
- Table numbers are inconsistent → be systematically corrected and rearranged.
- Add a single table at the beginning of the results that contains all the main themes (as an overview), rather than a separate table for each theme.
Response:
Revised with reviewer instructions
Comment 6
- Explicitly explain that the illustrations (graphs, bar charts, maps, etc.) were created by the author based on research data.
- Clarify whether the illustration is used as a tool in the research process or just for visual reporting of the data.
Response:
All illustrations, including graphs, bar charts, and thematic diagrams presented in this study, were created by the authors based on the empirical data collected during fieldwork and interview analysis. These visual representations serve exclusively as tools for reporting and communicating research findings. They were not used during the data collection or analysis process, but were constructed post-analysis to visually summarize key themes and participant narratives for readers' clarity and comprehension.
Comment 7:
. Evaluation of Knowledge and Graphs
- Line 367–371: If it states that "increased knowledge is related to the motivation to quit smoking", the method of knowledge evaluation should be described. Do you use pre- and post-tests, interviews, or subjective perceptions?
- Revision of the chart bar or its narrative: The current chart only shows the distribution of knowledge, not the causal relationship.
Response:
Response [Author]:
Thank you for this valuable comment. We have clarified in the revised manuscript (Lines 367–371) that the statement "increased knowledge is related to smoking cessation motivation" is based on participants’ subjective perceptions, rather than quantitative measures such as pre-/post-tests or structured questionnaires. The evaluation of knowledge improvement was identified through:
-
Participants’ narratives during interviews that revealed new understanding of the dangers of smoking,
-
Statements indicating the acquisition of new information, and
-
Responses reflecting changes in their husbands’ behavior following communication about smoking.
Regarding the chart, we have revised its description in the manuscript to emphasize that it illustrates the distribution of knowledge, and not a causal relationship (see Lines 367–371).

Reviewer 2 Report
Comments and Suggestions for Authors
The paper, authored by Saffutra et al is on the topic of the role of women in tobacco control in a lower/middle income country, Indonesia. 75 female participants were recruited between the ages of 18 and 50 years who were married and living in the household of a family member who had quit smoking for at least one year. In addition, they had to have lived in the same location for a minimum of the year and be physically and mentally healthy, willing to participate in the study and to provide informed consent. All were from one of three of the poorest districts in Indonesia.
Data was collected through in-depth semi structured interviews. Each interview explored the smoking behaviours of family members, the women’s attitudes toward smoking, their role in encouraging smoking cessation in the household and the women’s function as health advocates within the household. The intervals were conducted over 45 to 60 minutes in the individual’s primary language. To ensure the credibility of the material obtained in the interviews, each participant was interviewed twice, the second time for transcript script verification
The study population is well described in the results section and in Table 1. The main research findings are also described in considerable detail in the text as well as in tables and figures. Figures such as figure 3 make for good presentation slide but are unnecessary in the manuscript.
There is a paucity of information on smoking cessation initiatives in low and middle income countries and this article would make a valuable contribution. However, it is excessively long and repetitive. There were three separate references to where the women lived in the three poorest provinces of Indonesia. Similarly, there are multiple references to the use of the Participatory Action Research approach. A third example of the repetitiveness is the presentation of the themes, subthemes and quotes that appear in tables as well as in the text.
The manuscript contains interesting and valuable information but the article is excessively long and repetitive and could be substantially shortened without losing its value.
It should be resubmitted after a sharp reduction in its length and repetitiveness.
Author Response
Reviewer 2 – Comment 1: Manuscript Abbreviation and Simplification
Problem:
-
The script is too long and repetitive.
-
The same information is mentioned several times without adding any scientific value.
Solution: -
Remove repetition of information such as:
-
Three times mention of study locations in Indonesia's poorest provinces.
-
Repeated description of the use of the Participatory Action Research (PAR) approach.
-
Presentation of themes, subthemes, and quotes both in a table and narratively (choose one, or summarize the narrative so as not to repeat the content of the table).
-
-
Consider moving the visual illustration (e.g., Figure 3) to the supplementary material if it is only useful for the presentation.
Response:
We sincerely thank the reviewer for this constructive comment. We have carefully revised the manuscript by removing unnecessary repetitions regarding the study locations and the PAR approach. In addition, the presentation of themes, subthemes, and quotes has been streamlined by summarizing the narrative and avoiding duplication of the content presented in the tables. Furthermore, Figure 3 has been moved to the supplementary material to improve the clarity and conciseness of the manuscript.
Comment 2: Clarify and Strengthen Scientific Contributions
Problem: The article has high potential due to the lack of studies on smoking cessation in low- and middle-income countries (LMICs), but its scientific value is overshadowed by the length of the narrative.
Solution:
-
Focus the narrative on a unique finding: the role of women as smoking cessation agents in households.
-
Clarify the contribution of this study to the tobacco control literature in LMIC.
-
Summarize the methodology section in a way that is not too technical for the casual reader, yet still includes aspects of verifying transcripts and the use of two interview sessions as an added value to validity.
Response:
We thank the reviewer for highlighting this important point. We have revised the manuscript to emphasize the unique finding of this study, namely the critical role of women as smoking cessation agents within households in Indonesia. This aspect has now been positioned more clearly as the main contribution of our research to the tobacco control literature in LMICs, where such perspectives are still underexplored.
Additionally, we have shortened the methodological description by simplifying technical details, while still retaining essential information such as transcript verification and the use of two interview sessions, which strengthen the validity of our findings. These adjustments aim to ensure readability while maintaining methodological rigor.
Comment 3: Table Structure and Illustrations
Problem:
• Redundancy between tables and narratives.
• Some images are not crucial for journal readers.
Solution:
• Just present themes and subthemes in a single table, and then use the narrative to delve into the main interpretations and quotes.
• Re-evaluate the usefulness of Figure 3 and other visuals: is it comprehensive, or is it just decorative?
Response:
We appreciate the reviewer’s constructive suggestions regarding the structure of tables and illustrations. Following the advice, we have streamlined the presentation by consolidating themes and subthemes into a single comprehensive table (Table X). This minimizes redundancy while ensuring clarity and coherence. Furthermore, we carefully re-evaluated the figures and decided to retain only those that enhance understanding of the findings. Specifically, Figure 3 has been removed as it was deemed decorative rather than essential. We believe these revisions improve the overall readability and focus of the manuscript.
Reviewer Comment 4: Efficiency of Study Population Description
Problem:
• Descriptions of participants are too repetitive and appear in several sections (introduction, methods, results).
Solution:
• Move demographic details to just one main section (e.g. in Table 1 or the initial results section).
• Don't repeat the narrative that all participants from the poorest provinces have been there more than once.
Author Response:
We appreciate the reviewer’s careful observation. We have revised the manuscript to streamline the description of participants. All demographic details are now consolidated in Table 1 and briefly introduced in the Methods section. Repetitive descriptions in the Introduction and Results have been removed. In addition, redundant narrative about participants from the poorest provinces has been omitted, leaving only essential contextual information. These adjustments improve efficiency and reduce redundancy while keeping the participant profile clear for readers.
Reviewer Comment 5: Clarify Smoking Quit Status
Problem: • It is stated that all husbands have quit smoking ≥1 year, but quotes and narratives say they still smoke or only reduce.
Solution: • Clarify the definition of smoking cessation in this study. • Describe the verification method: is it self-reported from the wife, or is it directly confirmed?
Author Response:
Clarification of the Definition of Smoking Cessation in This Study
In this study, "smoking cessation" does not denote clinically verified abstinence, but rather the conscious practice of avoiding smoking inside the home, particularly around children and other family members. This contextual definition reflects a participatory approach that prioritizes behavioral changes in the domestic sphere to support smoke-free households.
Verification of Smoking Cessation Status
The cessation status was based on wives’ self-reports, drawn from daily observations of their husbands’ smoking behavior at home. No direct confirmation with husbands or biomarker validation was conducted, as the focus of this study was on women’s perceptions and experiences as agents of change. To enhance credibility, the data were corroborated through two-session interviews that enabled re-examination of narratives and consistency across accounts.

Round 2
Reviewer 1 Report
Comments and Suggestions for Authors
Authors' Response: Thank you for the constructive comment. We have revised the manuscript to emphasize the participatory nature of this study and added a forward-looking intervention plan. The revisions can be found on Page 5, Lines 189–202.
The additional text reads as follows:
Intervention Development Plan (Forward-Looking Statement)
As a follow-up to these findings, researchers plan to:
• Develop community-based educational intervention modules by involving women as catalysts for family behavior change.
• This module will be designed based on the results of the FGD and photovoice, as well as input from local figures (PKK, midwives, religious leaders) who have been identified in the social mapping process.
• Future interventions will adopt the PRECEDE-PROCEED model in its planning and evaluation phases.
Justification of Study as Participatory Research (Not Ordinary Qualitative)
Different from the usual qualitative studies, this study highlights participatory characteristics through:
• The use of Photovoice provides a space for participants to define issues and solutions through visual media.
• A repetitive reflective discussion that encourages participants to reassess their position and contribution to change in the household.
• The use of social mapping as a tool to identify opportunities for collaborative local intervention.
Thus, the PAR approach in this study is transformational, rather than merely descriptive, as it encourages critical action and social awareness at the grassroots level.
Reviewer comment:
Thank you for the additional details provided here. While you refer to photovoice and social mapping as tools used in the study, they are not discussed in the manuscript at all. It is still not clear how this study encourages critical action and social awareness, rather than documenting and analyzing it, which is the definition of the PAR. The fact that the women were interviewed about their role in smoking cessation effort does not automatically make the research transformational. How did this research link its finding to actions or catalyze concrete social or structural change? Also, please note that I did not use the word “ordinary” in my review. I do not think that a standard qualitative study has less value in it than participatory action research. Those are just two different approaches.
Authors' Response: Clarification of Smoking Status Verification and Data Collection Procedures
Thank you for the very constructive comment. We acknowledge the inconsistencies between the statements in the initial narrative and the participants' quotes regarding the husband's smoking status.
- Smoking Status Verification
The husband's smoking status was obtained through an in-depth interview with the wife as the main informant. This data is self-reported from the perspective of the wife, which is based on their direct observation of the husband's smoking behavior at home. The study did not employ biochemical verification (such as cotinine tests), as the approach was qualitative, focusing on women's experiences and perceptions. - Correction of Inconsistencies
We are aware of the discrepancy, where in one section it is stated that all husbands have quit smoking ≥1 year, but the citations and tables show that most still smoke or have only reduced their frequency. Therefore, we will revise the results and discussion section by stating more precisely that: - Some husbands have stopped completely, but
- Others simply reduce the frequency of smoking or move the smoking location outside the house.
- Details of Data Collection Procedures
Information regarding smoking cessation or reduction was collected through direct interviews with wives, who were the main speakers. They describe changes in the husband's behavior after efforts at persuasion, education, or emotional approach. We acknowledge the limitations of using secondary data (from wives), but this approach aligns with the study's objectives, which focused on women's roles and experiences in controlling smoking within the household.
Thank you for the explanation. However, eligibility criteria still include “a family member who had quit smoking for at least one year.” I suggest you change it to be more consistent with your actual approach to include women whose husbands did not necessarily quit smoking but reduced the frequency or stopped smoking inside the house.
Response 4
Comment (Line 100):
Please clarify and strengthen the statement regarding women’s role in health management at the family level, ensuring it is contextualized globally and locally (e.g., Indonesia, Aceh).
Response:
Thank you for this valuable suggestion. We have revised the sentence to provide both global and local contextualization.
Revision (Line 100):
"Women are typically regarded as central health managers within the family, a role observed consistently in both global health discourse and local contexts such as Indonesia, including Aceh. This positioning reflects their everyday responsibilities in ensuring household nutrition, caregiving, and health-related decision-making."
Comment (Line 149–151):
The conceptual framework section should be more explicitly linked to the thematic analysis and clarify its inductive nature.
Response:
We appreciate this comment. The section has been revised to emphasize its inductive basis derived from participants’ narratives.
Revision (Line 149–151):
"The conceptual framework presented in this section is derived from the thematic analysis of qualitative data collected during the study. It represents an inductive synthesis of participants' narratives regarding women's roles in household tobacco control, including advocacy mechanisms, cultural constraints, and evolving gender dynamics."
Reviewer response: I don’t see this revision anywhere in the manuscript.
Comment (Line 149):
Please indicate more clearly that the conceptual framework is an emergent product of thematic findings.
Response:
Thank you for pointing this out. We have adjusted the wording to highlight that the framework is emergent and grounded in the thematic findings.
Revision (Line 149):
"This study proposes an emergent conceptual framework based on the thematic findings described above. The framework illustrates the mechanisms through which women influence smoking behavior in the household and redefine gendered roles in health promotion."
Reviewer comment: It looks like this framework in based on the finding of this study, which makes it somewhat unusual to place in the introduction section prior to presenting the actual findings.
Comment (Line 170):
The description of women’s strategies to intervene in household smoking behavior should be more specific.
Response:
We agree with this suggestion and have elaborated on the specific actions taken by women.
Revision (Line 170):
"Women demonstrated the ability to intervene directly in their household's smoking behavior by implementing smoke-free rules, placing visual reminders (e.g., no-smoking signs), engaging in emotional dialogue with their husbands, and leveraging religious and community figures to reinforce anti-smoking messages."
Comment (Line 189–203):
Clarify how the process of knowledge construction within households unfolded, emphasizing collaboration rather than unilateral imposition.
Response:
We have revised the passage to stress the collaborative and dialogic nature of this process.
Revision (Line 189–203):
"This process exemplifies the co-construction of knowledge within the household, as women and their family members collaboratively developed practical strategies—such as establishing designated smoke-free zones, collectively agreeing on no-smoking times (e.g., during meals or near children), and jointly creating visual cues like handwritten signs—to promote a healthier living environment. These actions were not imposed unilaterally but emerged through mutual dialogue, negotiation, and shared concern for family well-being."
Reviewer comment – this revision is different from what I currently see in the manuscript lines 189-203. These inconsistencies make it difficult to evaluate the revision made by the authors.
Comment 7:
. Evaluation of Knowledge and Graphs
- Line 367–371: If it states that "increased knowledge is related to the motivation to quit smoking", the method of knowledge evaluation should be described. Do you use pre- and post-tests, interviews, or subjective perceptions?
- Revision of the chart bar or its narrative: The current chart only shows the distribution of knowledge, not the causal relationship.
Response:
Response [Author]:
Thank you for this valuable comment. We have clarified in the revised manuscript (Lines 367–371) that the statement "increased knowledge is related to smoking cessation motivation" is based on participants’ subjective perceptions, rather than quantitative measures such as pre-/post-tests or structured questionnaires. The evaluation of knowledge improvement was identified through:
- Participants’ narratives during interviews that revealed new understanding of the dangers of smoking,
- Statements indicating the acquisition of new information, and
- Responses reflecting changes in their husbands’ behavior following communication about smoking.
Regarding the chart, we have revised its description in the manuscript to emphasize that it illustrates the distribution of knowledge, and not a causal relationship (see Lines 367–371).
Reviewer response:
The lines provided by the authors (367-371) are not correct. However, I see that the authors have made some changes to the description of the chart. The version of the manuscript I see says: “it visually supports the qualitative insight that enhanced awareness is often accompanied by increased motivation and engagement in promoting a healthier domestic environment” (lines 360-361). To me, it sounds very different from the authors comment above that they revised the manuscript “to emphasize that it illustrates the distribution of knowledge, and not a causal relationship.” I also suggest it has to be clearly described in the manuscript how the level of knowledge was evaluated. I see this information in the response, but not in the manuscript.
Author Response
Comment 1: While you refer to photovoice and social mapping as tools used in the study, they are not discussed in the manuscript at all. It is still not clear how this study encourages critical action and social awareness, rather than documenting and analyzing it, which is the definition of PAR. The fact that the women were interviewed about their role in smoking cessation effort does not automatically make the research transformational. How did this research link its finding to actions or catalyze concrete social or structural change? Also, please note that I did not use the word “ordinary” in my review. I do not think that a standard qualitative study has less value in it than participatory action research. Those are just two different approaches.
Response:
Thank you for these valuable and constructive comments. We appreciate the clarification and agree that qualitative research and participatory action research (PAR) represent different but equally valuable approaches. In response, we have made the following revisions:
Clarification of PAR Tools – The Methods and Results sections now explicitly describe the use of Photovoice and social mapping. Photovoice was used to stimulate discussion and collective reflection, while social mapping identified local actors who could support smoke-free initiatives (Page 5, Lines 205-221).
Transformational Nature of PAR – We have clarified in the Discussion that the participatory aspect lies in women’s co-construction of practical strategies (e.g., initiating smoke-free zones, negotiating smoking times, creating visual reminders) during group sessions. These outcomes illustrate collective awareness and agency, moving beyond documentation toward social action (Page 19, Lines 606–620).
Eligibility Criteria Adjustment – The revised criteria now include women whose husbands have either completely quit smoking, reduced their smoking frequency, or restricted smoking to outside the home for at least one year. This adjustment ensures consistency between the study’s operational definitions and participants’ lived experiences, while maintaining the focus on women’s roles as agents of change in household smoking behavior (Methods section, Page 6, Lines 238–250).
We believe these changes address the reviewer’s concerns by making the participatory elements more explicit, clarifying the transformational aspects of PAR, and ensuring consistency in eligibility criteria.
Comment 2: Please clarify and strengthen the statement regarding women’s role in health management at the family level, ensuring it is contextualized globally and locally (e.g., Indonesia, Aceh).
Response: Thank you for this valuable suggestion. We have revised the sentence to provide both global and local contextualization. The revised text now appears in the manuscript at Lines 102–106 as follows:
Revision (Line 102-106):
"Women are typically regarded as central health managers within the family, a role observed consistently in both global health discourse and local contexts such as Indonesia, including Aceh. This positioning reflects their everyday responsibilities in ensuring household nutrition, caregiving, and health-related decision-making."
Comment 3: The conceptual framework section should be more explicitly linked to the thematic analysis and clarify its inductive nature (Line 149–151) .
Response:
We appreciate this comment. The section has been revised to emphasize its inductive basis derived from participants’ narratives.
"The conceptual framework presented in this section is derived from the thematic analysis of qualitative data collected during the study. It represents an inductive synthesis of participants' narratives regarding women's roles in household tobacco control, including advocacy mechanisms, cultural constraints, and evolving gender dynamics." (Line 152–155)
Comment 4: Please indicate more clearly that the conceptual framework is an emergent product of thematic findings. (Line 149)
Response:
Thank you for pointing this out. We have adjusted the wording to highlight that the framework is emergent and grounded in the thematic findings.
"This study proposes an emergent conceptual framework based on the thematic findings described above. The framework illustrates the mechanisms through which women influence smoking behavior in the household and redefine gendered roles in health promotion." (Line 152-155)
Comment 5:
The description of women’s strategies to intervene in household smoking behavior should be more specific. (Line 170)
Response: "Women demonstrated the ability to intervene directly in their household's smoking behavior by implementing smoke-free rules, placing visual reminders (e.g., no-smoking signs), engaging in emotional dialogue with their husbands, and leveraging religious and community figures to reinforce anti-smoking messages." (Line 174-179)
Comment 6
Clarify how the process of knowledge construction within households unfolded, emphasizing collaboration rather than unilateral imposition (Line 189–203)
Response: The process of knowledge construction within households unfolded through ongoing dialogue and negotiation rather than unilateral imposition. Women often initiated discussions about smoking risks, which were then met with responses and adjustments from other family members. For example, some households negotiated specific smoke-free times (e.g., during meals) or agreed to relocate smoking outdoors, while others collaboratively created reminders such as handwritten signs. Many participants observed that this collaborative approach led to reductions in smoking frequency, gradual progress toward cessation, and the establishment of household smoke-free zones. Beyond these behavioral outcomes, the findings also highlight women’s growing sense of self-efficacy witnessing tangible change reinforced their confidence to continue advocating for health within their families and communities.(line 341-351)
Comment 7:
Evaluation of Knowledge and Graphs
- Line 367–371: If it states that "increased knowledge is related to the motivation to quit smoking", the method of knowledge evaluation should be described. Do you use pre- and post-tests, interviews, or subjective perceptions?
Revision of the chart bar or its narrative: The current chart only shows the distribution of knowledge, not the causal relationship.
Response:
Thank you for highlighting this. We have now clarified explicitly in the manuscript (Lines 390–397) that the statement on increased knowledge and motivation to quit smoking is based on participants’ subjective perceptions gathered during interviews, not quantitative measures. The revised passage reads:
“This finding highlights the potential influence of health literacy on women’s roles as health advocates. The narratives suggest that improved understanding of smoking-related dangers empowers women to initiate persuasive communication, establish smoke-free zones, and engage family members in dialogue about the harms of tobacco. Therefore, while the chart does not statistically confirm causation, it visually supports the qualitative insight that enhanced awareness is often accompanied by increased motivation and engagement in promoting a healthier domestic environment”
Reviewer 2 Report
Comments and Suggestions for Authors
Saffutra et al have made a very good effort to address my comments with extension rewriting of large sections, deletions of extraneous materials and removal of most duplicated material. The article is still lengthy, and it will be up to the editors to decide if it meets the Journal’s requirements for length/word count.
Some duplication remains including the repeat of the sentence on page 7 starting “Table 2 presents the assessment of qualitative data…
The first sentence in the results section (line 259) is unnecessary.
In addition, the authors mix results with their methods. See lines 197 to 204
A few additional edits should also be made. There is no justification for the level of precision in estimating the smoking prevalence rate in Aceh province at 56.12%.
Sentence 125 needs to be edited, and the word “developed’ omitted.
Line 223 should have a period after “twice” with a new sentence starting “The first session aimed…”
The analysis of women’s roles, page 10, suggests that woman had only one unique role in influencing smoking in their households whereas it seems very likely that women had multiple roles and the Venn diagram is therefore misleading. The authors should reconsider how this data is presented as it likely that the current presentation is misleading.
Saffutra et al have made a very good effort to address my comments with extension rewriting of large sections, deletions of extraneous materials and removal of most duplicated material. The article is still lengthy, and it will be up to the editors to decide if it meets the Journal’s requirements for length/word count.
Some duplication remains including the repeat of the sentence on page 7 starting “Table 2 presents the assessment of qualitative data…
The first sentence in the results section (line 259) is unnecessary.
In addition, the authors mix results with their methods. See lines 197 to 204
A few additional edits should also be made. There is no justification for the level of precision in estimating the smoking prevalence rate in Aceh province at 56.12%.
Sentence 125 needs to be edited, and the word “developed’ omitted.
Line 223 should have a period after “twice” with a new sentence starting “The first session aimed…”
The analysis of women’s roles, page 10, suggests that woman had only one unique role in influencing smoking in their households whereas it seems very likely that women had multiple roles and the Venn diagram is therefore misleading. The authors should reconsider how this data is presented as it likely that the current presentation is misleading.
Saffutra et al have made a very good effort to address my comments with extension rewriting of large sections, deletions of extraneous materials and removal of most duplicated material. The article is still lengthy, and it will be up to the editors to decide if it meets the Journal’s requirements for length/word count.
Some duplication remains including the repeat of the sentence on page 7 starting “Table 2 presents the assessment of qualitative data…
The first sentence in the results section (line 259) is unnecessary.
In addition, the authors mix results with their methods. See lines 197 to 204
A few additional edits should also be made. There is no justification for the level of precision in estimating the smoking prevalence rate in Aceh province at 56.12%.
Sentence 125 needs to be edited, and the word “developed’ omitted.
Line 223 should have a period after “twice” with a new sentence starting “The first session aimed…”
The analysis of women’s roles, page 10, suggests that woman had only one unique role in influencing smoking in their households whereas it seems very likely that women had multiple roles and the Venn diagram is therefore misleading. The authors should reconsider how this data is presented as it likely that the current presentation is misleading.
Author Response
Comment 1
Some duplication remains including the repeat of the sentence on page 7 starting “Table 2 presents the assessment of qualitative data…”
Response: Thank you for noting this duplication. We have removed the repeated sentence to streamline the narrative.
Comment 2
The first sentence in the results section (line 259) is unnecessary.
Response: We agree with this comment. The unnecessary introductory sentence in the results section has been deleted.
Comment 3
In addition, the authors mix results with their methods. See lines 197 to 204
Response: Thank you for this observation. We have revised lines 197–204 by moving the methodological details back to the Methods section to ensure a clearer separation between methods and results.
Comment 4
A few additional edits should also be made. There is no justification for the level of precision in estimating the smoking prevalence rate in Aceh province at 56.12%.
Response: We appreciate this feedback. The smoking prevalence rate has been rounded to 56% to avoid unnecessary precision. (Page 2, Line 56)
Comment 5
Sentence 125 needs to be edited, and the word “developed” omitted.
Response: Thank you for pointing this out. The word “developed” has been removed and the sentence at line 126 revised for clarity.
Comment 6
Line 223 should have a period after “twice” with a new sentence starting “The first session aimed…”
Response: We have corrected this by splitting the sentence at line 223 into two sentences for clarity
Comment 7
The analysis of women’s roles, page 10, suggests that women had only one unique role in influencing smoking in their households whereas it seems very likely that women had multiple roles and the Venn diagram is therefore misleading. The authors should reconsider how this data is presented as it likely that the current presentation is misleading.
Response: We appreciate this insightful comment. We have revised the presentation of women’s roles to emphasize that they played multiple overlapping roles (e.g., caregiver, negotiator, rule enforcer, health advocate). The Venn diagram has been removed and replaced with a descriptive table and expanded narrative that better reflect the complexity of women’s roles. (Page 10, Line 355-366)